# Learning Invariances using the Marginal Likelihood

**Mark van der Wilk**
PROWLER.io
Cambridge, UK
mark@prowler.io

**Matthias Bauer**
MPI for Intelligent Systems
University of Cambridge
msb55@cam.ac.uk

**ST John**
PROWLER.io
Cambridge, UK
st@prowler.io

**James Hensman**
PROWLER.io
Cambridge, UK
james@prowler.io

## Abstract

Generalising well in supervised learning tasks relies on correctly extrapolating the training data to a large region of the input space. One way to achieve this is to constrain the predictions to be invariant to transformations of the input that are known to be irrelevant (e.g. translation). Commonly, this is done through data augmentation, where the training set is enlarged by applying hand-crafted transformations to the inputs. We argue that invariances should instead be incorporated in the model structure, and learned using the *marginal likelihood*, which correctly rewards the reduced complexity of invariant models. We demonstrate this for Gaussian process models, due to the ease with which their marginal likelihood can be estimated. Our main contribution is a variational inference scheme for Gaussian processes containing invariances described by a sampling procedure. We learn the sampling procedure by backpropagating through it to maximise the marginal likelihood.

## 1 Introduction

In supervised learning, we want to predict some quantity $y \in \mathcal{Y}$ given an input $\mathbf{x} \in \mathcal{X}$ from a limited number of $N$ training examples $\{\mathbf{x}_n, y_n\}_{n=1}^N$. We want our model to make correct predictions in as much of the input space $\mathcal{X}$ as possible. By constraining our predictor to make similar predictions for inputs which are modified in ways that are irrelevant to the prediction (e.g. small translations, rotations, or deformations for handwritten digits), we can generalise what we learn from a single training example to a wide range of new inputs. It is common to encourage these *invariances* by training on a dataset that is enlarged by training examples that have undergone modifications that are known to not influence the output – a technique known as *data augmentation*. Developing an augmentation for a particular dataset relies on expert knowledge, trial and error, and cross-validation. This human input makes data augmentation undesirable from a machine learning perspective, akin to hand-crafting features. It is also unsatisfactory from a Bayesian perspective, according to which assumptions and expert knowledge should be explicitly encoded in the prior distribution only. By adding data that are not true observations, the posterior may become overconfident, and the marginal likelihood can no longer be used to compare to other models.

In this work, we argue that data augmentation should be formulated as an invariance in the functions that are learnable by the model. To do so, we investigate prior distributions which incorporate invariances. The main benefit of treating invariances in this way is that models with different invariances can be compared using the marginal likelihood. As a consequence, parameterised invariances can even be learned by backpropagating through the marginal likelihood.

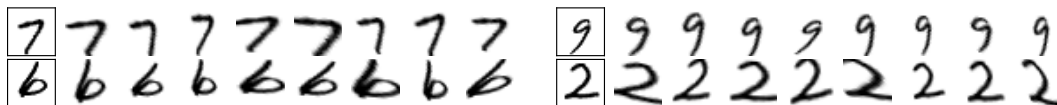

Figure 1: Samples describing the *learned invariance* for some example MNIST digits (squares). The method becomes insensitive to the rotations, shears, and rotations that are present in the samples.

We start from Gaussian process (GP) approximations, as they provide high-quality marginal likelihood approximations. We build on earlier work by developing a practical variational inference scheme for Gaussian process models that have invariances built into the kernel. Our approach overcomes the major technical obstacle that our invariant kernels cannot be computed in closed form, which has been a requirement for kernel methods until now. Instead, we only require unbiased estimates of the kernel for learning the GP and its hyperparameters. The estimate is constructed by sampling from a distribution that characterises the invariance (fig. 1).

We believe this work to be exciting, as it simultaneously provides a Bayesian alternative to data augmentation, and a natural method for learning invariances in a supervised manner. Additionally, the ability to use kernels that do not admit closed-form evaluation may be of use for the kernel community in general, as it may open the door to new kernels with interesting properties beyond invariance.

## 2 Related work

Incorporating invariances into machine learning models is commonplace, and has been addressed in many ways over the years. Despite the wide variety of methods for incorporating *given* invariances, there are few solutions for learning which invariances to use. Our approach is unique in that it performs direct end-to-end training using a supervised objective function. Here we present a brief review of existing methods, grouped into three rough categories.

**Data augmentation.** As discussed, data augmentation refers to creating additional training examples by transforming training inputs in ways that do not change the prediction [Beymer and Poggio, 1995; Niyogi et al., 1998]. The larger dataset constrains the model's predictions to be correct for a larger region of the input space. For example, Loosli et al. [2007] propose augmenting with small rotations, scaling, thickening/thinning and deformations. On modern deep learning tasks like ImageNet [Deng et al., 2009], it is standard to apply flips, crops, and colour alterations [Krizhevsky et al., 2012; He et al., 2016]. Most attempts at learning the data augmentation focus on generating more data from unsupervised models trained on the inputs. Hauberg et al. [2016] learn a distribution of mappings that transform between pairs of input images, which are then sampled and applied to random training images, while Antoniou et al. [2017] use a GAN to capture the input density.

**Model constraints.** An alternative to letting a flexible model learn an invariance described by a data augmentation is to constrain the model to exhibit invariances through clever parameterisation. Convolutional Neural Networks (CNNs) [LeCun et al., 1989, 1998] are a ubiquitous example of this, and work by applying the same filters across different image locations, giving a form of translation invariance. Cohen and Welling [2016] extend this with filters that are shared across other transformations like rotations. Invariances have also been incorporated into kernel methods. MacKay [1998] introduced the periodic kernel for functions invariant to shifts by the period. More sophisticated invariances suitable for images, like translation invariance, were discussed by Kondor [2008]. Ginsbourger et al. [2012, 2013, 2016] investigated similar kernels in the context of Gaussian processes. More recently, van der Wilk et al. [2017] introduced a Gaussian process with generalisation properties similar to CNNs, together with a tractable approximation. The same method can also be used to improve the scaling of the invariant kernels introduced by the earlier papers, and our method is based on it. For similar kernels, Raj et al. [2017] present a random feature based model approximation for invariances that are not learned. A final approach is to map the inputs to some fundamental space which is constant for all inputs that we want to be invariant to [Kondor, 2008; Ginsbourger et al., 2012]. For example, we can achieve rotational invariance by mapping the input image to some canonical rotation, on which classification is then performed. Jaderberg et al. [2015] do this by learning to "untransform" digits to a canonical orientation before performing classification.

**Regularisation.** Instead of placing hard constraints on the functions that can be represented, regularisation encourages desired solutions by adding extra penalties to the objective function. Simard et al. [1992] encourage locally invariant solutions by penalising the derivative of the classifier in the directions that the model should be invariant to. SVMs have also been regularised to encourage invariance to local perturbations, notably in Schölkopf et al. [1998], Chapelle and Schölkopf [2002], and Graepel and Herbrich [2004].

## 3 The influence of invariance on the marginal likelihood

In this work, we aim to improve the generalisation ability of a function $f : \mathcal{X} \to \mathcal{Y}$ by constraining it to be invariant. By following the Bayesian approach and making the invariance part of the prior on $f(\cdot)$, we can use the marginal likelihood to learn the correct invariances in a supervised manner. In

this section we will justify our approach, first by defining invariance, and then by showing why the marginal likelihood, rather than the "regular" likelihood, is a natural objective for learning.

## 3.1 Invariance

In this work we will distinguish between what we will refer to as "strict invariance" and "insensitivity". For the definition of strict invariance we follow the standard definition that is also used by Kondor [2008, §4.4] and Ginsbourger et al. [2012], where we require the value of our function $f(\cdot)$ to remain unchanged if any transformation $t : \mathcal{X} \to \mathcal{X}$ from a set $\mathcal{T}$ is applied to the input:

$$f(\mathbf{x}) = f(t(\mathbf{x})) \qquad \forall \mathbf{x} \in \mathcal{X} \qquad \forall t \in \mathcal{T} . \tag{1}$$

The set of all transformations $\mathcal{T}$ determines the invariance. For example, $\mathcal{T}$ would be the set of all rotations if we want $f(\cdot)$ to be rotationally invariant.

For many tasks, imposing strict invariance is too restrictive. For example, imposing rotational invariance will likely help the recognition of handwritten 2s, especially if they are presented in a haphazardly rotated way, while this same invariance may be detrimental for telling apart 6s and 9s in their natural orientation. For this reason, our main focus in this paper is on approximate invariances, where we want our function to not change "too much" after transformations on the input. We call this notion of invariance *insensitivity*. This notion is actually the most common in the related work, with data augmentation and regularisation approaches only enforcing $f(\cdot)$ to take similar values for transformed inputs, rather than *exactly* the same value. In this work we formalise insensitivity as controlling the probability of a large deviation in $f(\cdot)$ after applying a random transformation $t \in \mathcal{T}$ to the input:

$$P\Big([f(\mathbf{x}) - f(t(\mathbf{x}))]^2 > L\Big) < \epsilon \qquad \forall \mathbf{x} \in \mathcal{X} \qquad t \sim p(t) . \tag{2}$$

When working with insensitivity in practice, we conceptually think about the distribution of points that are generated by the transformations, rather than the transformations themselves. This gives a formulation that is much closer to the aim of data augmentation: a limit on the variation of the function for augmented points. Writing the distribution of points $\mathbf{x}_a$ obtained from applying the transformations as $p(\mathbf{x}_a|\mathbf{x})$, we can instead write:

$$P\Big([f(\mathbf{x}) - f(\mathbf{x}_a)]^2 > L\Big) < \epsilon \qquad \forall \mathbf{x} \in \mathcal{X} \qquad \mathbf{x}_a \sim p(\mathbf{x}_a|\mathbf{x}) . \tag{3}$$

For the remainder of this paper we will refer to both strict invariance and insensitivity as simply "invariance". Our method treats both similarly, with strict invariance being a special case of insensitivity.

From these definitions, we can see how these constraints can improve generalisation. While the prediction of a non-invariant learning method is only influenced in a small region around a training point, invariant models are constrained to make similar predictions in a much larger area, with the area being determined by the set or distribution of transformations. Insensitivity is particularly useful, as it allows local invariances. Making $f(\cdot)$ insensitive to rotation can help the classification of 6s that have been rotated by small angles, while also allowing $f(\cdot)$ to give a different prediction for 9s.

## 3.2 Marginal likelihood

Most current machine learning models are trained by maximising the regularised likelihood $p(\mathbf{y}|f(\cdot))$ with respect to parameters of the regression function $f(\cdot)$. One issue with this objective function is that it does not distinguish between models which fit the training data equally well, but will have different generalisation characteristics. We see an example of this in fig. 2. The figure shows data from a function that is invariant to switching the two input coordinates. We can train a model with the invariance embedded into the prior, and a non-invariant model. In terms of RMSE on the training set (which is proportional to the log likelihood), both models fit the training data very well, with the non-invariant model even fitting marginally better. However, the invariant model generalises much better, as points on one half of the input inform the function on the other half.

The marginal likelihood is found by integrating the likelihood $p(\mathbf{y}|f)$ over the prior on $f(\cdot)$,

$$p(\mathbf{y}|\theta) = \int p(\mathbf{y}|f)p(f|\theta)\mathrm{d}f , \tag{4}$$

and effectively captures the model complexity as well as the data fit [Rasmussen and Ghahramani, 2001; MacKay, 2002; Rasmussen and Williams, 2005]. It is also closely related to bounds on the generalisation error [Seeger, 2003; Germain et al., 2016]. Our example in fig. 2 also shows that the marginal likelihood *does* correctly identify the invariant model as the one that generalises best.

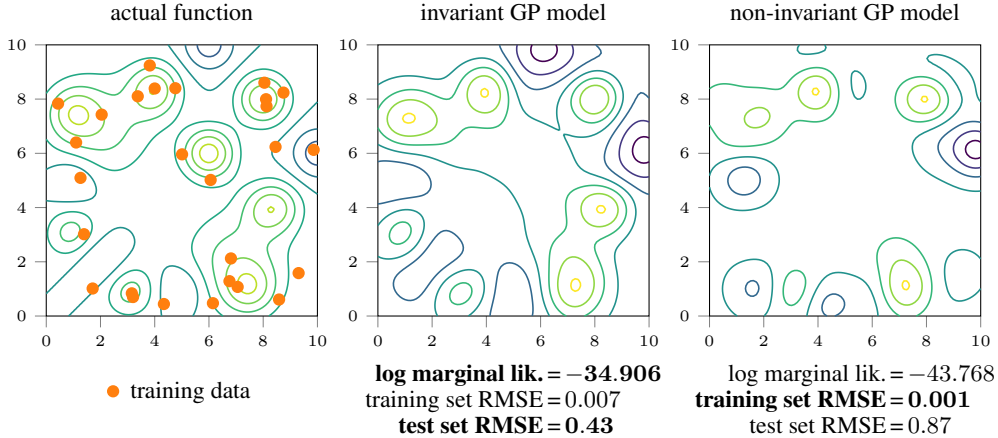

Figure 2: Data from a symmetric function *(left)* with the solutions learned by invariant *(middle)* and non-invariant *(right)* Gaussian processes. While the non-invariant model fits better to the training data, the invariant model generalises better. The marginal likelihood identifies the best model.

To understand how the invariance affects the marginal likelihood, and why a high marginal likelihood can indicate good generalisation performance, we decompose it using the product rule of probability and by splitting up our dataset $\mathbf{y}$ into chunks $\{\mathbf{y}_1, \mathbf{y}_2, \ldots, \mathbf{y}_C\}$:

$$p(\mathbf{y}|\theta) = p(\mathbf{y}_1|\theta)p(\mathbf{y}_2|\mathbf{y}_1,\theta)p(\mathbf{y}_3|\mathbf{y}_{1:2},\theta) \prod_{c=4}^{C} p(\mathbf{y}_c|\mathbf{y}_{1:c-1},\theta). \tag{5}$$

From this we see that the marginal likelihood measures how well previous chunks of data predict future ones. It seems reasonable that if previous chunks of the training set accurately predict later ones, that our entire training set will predict well on a test set as well. We can apply this insight to the example in fig. 2 by dividing the training set into chunks consisting of the points in the top left, and the bottom right halves. The non-invariant model only generalises locally, and will therefore make predictions close to the prior for the opposing half. The invariant model is constrained to predict exactly the same for the opposing half as it has learned for the observed half, and will therefore be confident *and* correct, giving a much higher marginal likelihood. Note that if the invariance had been detrimental to predictive performance (e.g. if $f(\cdot)$ was constrained to be anti-symmetric) the marginal likelihood would have been poor, as the model would have made incorrect predictions for $\mathbf{y}_2$.

Given that the marginal likelihood correctly identifies which invariances in the prior benefit generalisation, we focus our efforts in the rest of this paper on finding a good approximation that can be practically used to learn invariances.

## 4 Inference for Gaussian processes with invariances

In the previous section we argued that the marginal likelihood was a more appropriate objective function for learning invariances than the regular likelihood. Marginal likelihoods are commonly hard to compute, but good approximations exist for Gaussian processes[1] with simple kernels. In this section, we focus our efforts on Gaussian processes based on kernels with complex, parameterised invariances built in, for which we will derive a practical marginal likelihood approximation.

Our approximation is based on earlier variational lower bounds for Gaussian processes. While Turner and Sahani [2011] point out that variational bounds do introduce bias to hyperparameter estimates, the bias is well-understood in our case, and is reduced by using sufficiently many inducing points [Bauer et al., 2016]. In the literature, this method is commonplace for both regression and classification tasks [Titsias, 2009; Hensman et al., 2013, 2015a; van der Wilk et al., 2017; Kim and Teh, 2018].

### 4.1 Invariant Gaussian processes

Our starting point is the double-sum construction for priors over strictly invariant functions [Kondor, 2008; Ginsbourger et al., 2012], which we briefly review. If $f(\cdot)$ is strictly invariant to a set of

transformations, $f(\cdot)$ must also be invariant to compositions of transformations, as the same invariance holds at the transformed point $t(\mathbf{x})$. The set of all compositions of transformations forms the *group* $G$. The set of all points that can be obtained by applying transformations in $G$ to a point $\mathbf{x}$ forms the *orbit* of $\mathbf{x}$: $\mathcal{A}(\mathbf{x}) = \{t(\mathbf{x}) \mid t \in G\}$. We use $P$ to denote the number of elements in $\mathcal{A}(\mathbf{x})$. All input points which can be transformed into one another by an element of $G$ share an orbit, and must also have the same function value. This implies a simple construction of a strictly invariant function $f(\cdot)$ from a non-invariant function $g(\cdot)$. We can simply sum $g(\cdot)$ over the orbit of a point:

$$f(\mathbf{x}) = \sum_{\mathbf{x}_a \in \mathcal{A}(\mathbf{x})} g(\mathbf{x}_a) \,. \tag{6}$$

By placing a GP prior on $g(\cdot) \sim \mathcal{GP}(0, k_g(\cdot, \cdot))$, we imply a GP on invariant functions $f(\cdot)$, since Gaussians are closed under summation. We find that the prior on $f(\cdot)$ has a double-sum kernel:

$$k_f(\mathbf{x}, \mathbf{x}') = \mathbb{E}_g \left[ \sum_{\mathbf{x}_a \in \mathcal{A}(\mathbf{x})} g(\mathbf{x}_a) \sum_{\mathbf{x}'_a \in \mathcal{A}(\mathbf{x}')} g(\mathbf{x}'_a) \right] = \sum_{\mathbf{x}_a \in \mathcal{A}(\mathbf{x})} \sum_{\mathbf{x}'_a \in \mathcal{A}(\mathbf{x}')} k_g(\mathbf{x}_a, \mathbf{x}'_a) \,. \tag{7}$$

Earlier we argued that insensitivity is often more desirable. In order to relax the constraint of strict invariance, we relax the constraint that we sum over an orbit. Instead, we consider $\mathcal{A}(\mathbf{x})$ to be an arbitrary set of points, which we will refer to as an *augmentation set*, describing what input changes $f(\cdot)$ should be insensitive to. If two inputs have significantly overlapping augmentation sets, then their corresponding function values are constrained to be similar, as many terms in the sum of eq. (6) are shared (see appendix A for how this achieves insensitivity in the sense of eq. (2)). This kernel was also studied by Dao et al. [2018] as a first-order approximation of data augmentation, and Raj et al. [2017] as a local invariance.

We can also consider infinite augmentation sets, where we describe the relative density of elements using a probability density, which we refer to as the augmentation density $p(\mathbf{x}_a|\mathbf{x})$. We will take $p(\mathbf{x}_a|\mathbf{x})$ to be a process which perturbs the training data, much like how data augmentation is performed. Following a similar argument to the above, this results in a kernel that is doubly integrated over the augmentation distribution $p(\mathbf{x}_a|\mathbf{x})$:

$$k_f(\mathbf{x}, \mathbf{x}') = \iint p(\mathbf{x}_a|\mathbf{x}) p(\mathbf{x}'_a|\mathbf{x}') k_g(\mathbf{x}_a, \mathbf{x}'_a) \mathrm{d}\mathbf{x}_a \mathrm{d}\mathbf{x}'_a \,. \tag{8}$$

We collect all the parameters of the kernel, consisting of the parameters of the augmentation distribution and the base kernel, in $\theta$ (dropped from notation for brevity), and treat them as hyperparameters of the model, which we will tune using an approximation to the marginal likelihood.

When using these kernels, we face an additional obstacle on top of the usual problems with scalability and non-conjugate inference. The sums over large orbits prohibit the evaluation of eq. (7), while the integrals in eq. (8) are analytically intractable for interesting invariances. Over the next few sections, we develop an approximation that will allow us to evaluate a lower bound to the marginal likelihood which only requires samples of the orbit $\mathcal{A}(\mathbf{x})$ or augmentation distribution $p(\mathbf{x}_a|\mathbf{x})$. This allows us to minibatch not only over examples in the training set, but also over samples that describe the desired invariances.

## 4.2 Variational inference using inducing points

We want to use the invariant GP defined in the previous section as a prior over functions for regression and classification models. With slight abuse of notation we denote our prior $p(f)$, which will be Gaussian for the marginal of any finite set of input points. We denote sets of inputs as matrices $\mathbf{X} \in \mathbb{R}^{N \times D}$, and observations as vectors $\mathbf{f} = \{f(\mathbf{x}_n)\}_{n=1}^N = f(\mathbf{X})$. We denote our model:

$$f \mid \theta \sim \mathcal{GP}(0, k_f(\cdot, \cdot)) \,, \qquad\qquad y_n \mid f, \mathbf{x}_n \overset{iid}{\sim} p(y_n \mid f(\mathbf{x}_n)) \,, \tag{9}$$

where $p(y_i|f(\mathbf{x}_i))$ is a Gaussian likelihood for regression, Bernoulli for binary classification, etc. The marginal of $p(f)$ for a finite number of observations is a Gaussian distribution with covariance $\mathbf{K}_{\mathbf{ff}}$:

$$p(f(\mathbf{X})) = p(\mathbf{f}) = \mathcal{N}(0, \mathbf{K}_{\mathbf{ff}}) \,, \qquad\qquad [\mathbf{K}_{\mathbf{ff}}]_{nn'} = k_f(\mathbf{x}_n, \mathbf{x}_{n'}) \,. \tag{10}$$

Inference in GPs suffers from two main difficulties. First, inference is only analytically tractable for Gaussian likelihoods. Second, computation in GP models is well-known to scale badly with the

dataset size, requiring $\mathcal{O}(N^3)$ computations on $\mathbf{K_{ff}}$. Approximate inference using *inducing variables* [Quiñonero-Candela and Rasmussen, 2005] can be used to address both of these problems. We follow the variational approach (referring to Titsias [2009]; Hensman et al. [2013, 2015b] for the full details) by introducing an approximate Gaussian process posterior denoted $q(f)$, which is constructed by conditioning on $M < N$ "inducing observations". The shape of $q(f)$ can be adjusted by changing the input locations $\{\mathbf{z}_m\}_{m=1}^M = \mathbf{Z}$ and output mean $\mathbf{m}$ and covariance $\mathbf{S}$ of these inducing observations. This results in an approximate posterior of the form $q(f) = \mathcal{GP}(\mu(\cdot), \nu(\cdot, \cdot))$ with

$$\mu(\cdot) = \mathbf{k_u^\intercal}(\cdot)\mathbf{K_{uu}^{-1}}\mathbf{m}, \qquad \nu(\cdot, \cdot) = k(\cdot, \cdot) - \mathbf{k_u^\intercal}(\cdot)\mathbf{K_{uu}^{-1}}[\mathbf{K_{uu}} - \mathbf{S}]\mathbf{K_{uu}^{-1}}\mathbf{k_u}(\cdot), \qquad (11)$$

where $[\mathbf{K_{uu}}]_{mm'} = k(\mathbf{z}_m, \mathbf{z}_{m'})$ (analogous to $\mathbf{K_{ff}}$ only using the inducing input locations $\mathbf{Z}$), and $\mathbf{k_u}(\cdot) = [k(\mathbf{z}_m, \cdot)]_{m=1}^M$ is the covariance between the inducing outputs and the rest of the process.

We select the variational parameters by numerical maximisation of the marginal likelihood lower bound (or, evidence lower bound: ELBO) using stochastic optimisation [Hensman et al., 2013]. This correctly minimises the KL divergence between the approximate and exact posteriors $\mathrm{KL}[q(f)||p(f|\mathbf{y})]$ [Matthews et al., 2016]. The ELBO is given by

$$\log p(\mathbf{y}) \geq \mathcal{L} = \sum_{n=1}^{N} \mathbb{E}_{q(f(\mathbf{x}_n))}[\log p(\mathbf{y}|f(\mathbf{x}_n)] - \mathrm{KL}[q(\mathbf{u})||p(\mathbf{u})]. \qquad (12)$$

We find the expectation under $q(f(\mathbf{x}_n))$ either analytically or by Monte Carlo. In order to reduce the cost of evaluating the whole sum, we evaluate the bound stochastically by sub-sampling. This technique allows Gaussian processes to be applied to large datasets with general likelihoods. However, it does not address the issue of needing to evaluate our intractable kernel $k_f$ (eqs. (7) and (8)).

### 4.3 Inter-domain inducing variables

The problem of evaluating large double sums was encountered by van der Wilk et al. [2017] for convolutional and strictly invariant kernels. Their solution was based on the observation that problems with evaluating the bound (eq. (12)) stemmed from intractabilities in the approximate posterior $q(f)$, since this is where the kernel evaluations are needed. By choosing a different parameterisation of $q(f)$, the cost of evaluating the approximate posterior for a minibatch of $\tilde{N}$ points can be reduced from $\mathcal{O}(\tilde{N}^2 + (\tilde{N}M + M^2)P^2 + M^3)$ to $\mathcal{O}(\tilde{N}P^2 + \tilde{N}MP + M^2 + M^3)$ – a significant saving, particularly when $\tilde{N}$ is small, and $M$ and $P$ are large.

This can be achieved simply by constructing the posterior from inducing variables in $g(\cdot)$ instead of $f(\cdot)$. Approximations constructed using observations in different spaces are said to use *inter-domain* inducing variables [Figueiras-Vidal and Lázaro-Gredilla, 2009], and can use the same variational bound as in eq. (12) [Matthews et al., 2016], with only $\mathbf{K_{uu}}$ and $\mathbf{k_u}(\cdot)$ being modified in $q(f(\mathbf{x}_n))$:

$$k_{\mathbf{fu}}(\mathbf{x}, \mathbf{z}) = \mathbb{E}_{p(g)}[f(\mathbf{x})g(\mathbf{z})] = \sum_{\mathbf{x}_a \in \mathcal{A}(\mathbf{x})} k(\mathbf{x}_a, \mathbf{z}), \qquad k_{\mathbf{uu}}(\mathbf{z}, \mathbf{z}') = k_g(\mathbf{z}, \mathbf{z}'). \qquad (13)$$

The new covariances require only a single sum, or no sum at all. Only $k_f(\mathbf{x}_n, \mathbf{x}_n)$ still requires a double sum, although this can be mitigated by keeping the minibatch size $\tilde{N}$ small.

While this technique allows variational inference to be applied to invariant kernels with moderately sized orbits, similar to the convolutional kernels in van der Wilk et al. [2017], it does not help when even a single sum is too large. This technique is not applicable when intractable integrals appear (e.g. eq. (8)), since evaluations of the intractable $k_f$ are still needed, and the covariance function $k_{\mathbf{fu}}$ also requires an intractable integral:

$$k_{\mathbf{fu}}(\mathbf{x}, \mathbf{z}) = \mathbb{E}_{p(g)}\left[\int p(\mathbf{x}_a|\mathbf{x})g(\mathbf{x}_a)g(\mathbf{z})\right]\mathrm{d}\mathbf{x}_a = \int p(\mathbf{x}_a|\mathbf{x})k(\mathbf{x}_a, \mathbf{x})\mathrm{d}\mathbf{x}_a. \qquad (14)$$

### 4.4 An estimator using samples describing invariances

We now show that the inter-domain parameterisation above allows us to, for certain likelihoods, create an unbiased estimator of the lower bound in eq. (12) using unbiased estimates of $k_f$ and $k_{\mathbf{fu}}$. We start with discussing the estimator for the Gaussian likelihood here, leaving some non-Gaussian likelihoods for the next section. We only consider the integral formulation of the kernel, as in eq. (8), as sub-sampling of augmentation sets requires only a minor tweak[2].

For Gaussian likelihoods, the expectation in eq. (12) can be evaluated analytically, giving the bound

$$\mathcal{L} = \sum_{n=1}^{N} \left[ -\log 2\pi\sigma^2 - \frac{1}{2\sigma^2}\left(y_n^2 - 2y_n\mu_n + \mu_n^2 + \sigma_n^2\right)\right] - \mathrm{KL}[q(\mathbf{u})||p(\mathbf{u})]\,, \qquad (15)$$

with $\mu_n = \mu(\mathbf{x}_n)$ and $\sigma_n^2 = \nu(\mathbf{x}_n, \mathbf{x}_n)$ (eq. (11)) being the only terms which depend on the intractable $k_f$ and $k_{\mathbf{fu}}$ covariances. The KL term is tractable, as it only depends on $\mathbf{K_{uu}}$, which can be evaluated from $k_g$ directly (eq. (13)).

We aim to construct unbiased estimators $\widehat{\mu_n}$, $\widehat{\mu_n^2}$ and $\widehat{\sigma_n^2}$ for the intractable terms, which only rely on samples of $p(\mathbf{x}_a|\mathbf{x})$. The posterior mean can be estimated easily using a Monte Carlo estimate of $k_{\mathbf{fu}}$:

$$\mu_n = \mathbf{k_{f_n u}}\mathbf{K_{uu}^{-1}}\mathbf{m} = \left[\int p(\mathbf{x}_a|\mathbf{x}_n)\mathbf{k}_g(\mathbf{x}_a, \mathbf{Z})\mathrm{d}\mathbf{x}_a\right]\mathbf{K_{uu}^{-1}}\mathbf{m} \qquad \implies \qquad \widehat{\mu_n} = \widehat{\mathbf{k}}_{\mathbf{f_n u}}\mathbf{K_{uu}^{-1}}\mathbf{m}\,, \quad (16)$$

$$\hat{k}_{\mathbf{fu}}(\mathbf{x}, \mathbf{z}) = \sum_{s=1}^{S} k_g(\mathbf{x}^{(s)}, \mathbf{z}), \qquad \mathbf{x}^{(s)} \sim p(\mathbf{x}_a|\mathbf{x}). \qquad (17)$$

For $\mu_n^2$ and $\sigma_n^2$, we start by rewriting them so we can focus on estimators of $k_f(\mathbf{x}_n, \mathbf{x}_n)$ and $\mathbf{k}_{\mathbf{f_n u}}^{\mathsf{T}}\mathbf{k}_{\mathbf{f_n u}}$:

$$\mu_n^2 = \mathbf{k_{f_n u}}\mathbf{K_{uu}^{-1}}\mathbf{m}\mathbf{m}^{\mathsf{T}}\mathbf{K_{uu}^{-1}}\mathbf{k_{f_n u}^{\mathsf{T}}} = \mathrm{Tr}\left[\mathbf{K_{uu}^{-1}}\mathbf{m}\mathbf{m}^{\mathsf{T}}\mathbf{K_{uu}^{-1}}\left(\mathbf{k_{f_n u}^{\mathsf{T}}}\mathbf{k_{f_n u}}\right)\right]\,, \qquad (18)$$

$$\sigma_n^2 = k_f(\mathbf{x}_n, \mathbf{x}_n) - \mathrm{Tr}\left[\mathbf{K_{uu}^{-1}}(\mathbf{K_{uu}} - \mathbf{S})\mathbf{K_{uu}^{-1}}\left(\mathbf{k_{f_n u}^{\mathsf{T}}}\mathbf{k_{f_n u}}\right)\right]\,. \qquad (19)$$

We treat $k_f(\mathbf{x}_n, \mathbf{x}_n)$ and an element of $\mathbf{k}_{\mathbf{f_n u}}^{\mathsf{T}}\mathbf{k}_{\mathbf{f_n u}}$ identically, as they can both be written as the integral

$$I = \iint p(\mathbf{x}_a|\mathbf{x}_n)p(\mathbf{x}_a'|\mathbf{x}_n)r(\mathbf{x}_a, \mathbf{x}_a')\mathrm{d}\mathbf{x}_a\mathrm{d}\mathbf{x}_a'\,, \qquad (20)$$

with $r = k_g(\mathbf{x}_a, \mathbf{x}_a')$ and $r = k_{\mathbf{fu}}(\mathbf{x}_a, \mathbf{z}_m)k_{\mathbf{fu}}(\mathbf{x}_a', \mathbf{z}_{m'})$, respectively. A simple Monte Carlo estimate would require sampling two independent sets of points for $\mathbf{x}_a$ and $\mathbf{x}_a'$. We would like to sample only a single one, so all the necessary quantities can be estimated with the same "minibatch" of augmented points. We propose to use the following estimator, which we show to be unbiased in appendix B.

$$\hat{I} = \frac{1}{S(S-1)}\sum_{s=1}^{S}\sum_{s'=1}^{S} r\left(\mathbf{x}^{(s)}, \mathbf{x}^{(s')}\right)(1 - \delta_{ss'})\,, \qquad \mathbf{x}^{(s)} \sim p(\mathbf{x}_a|\mathbf{x})\,. \qquad (21)$$

We now have unbiased estimates for all quantities needed to optimise the variational lower bound.

### 4.5   Logistic classification with Pòlya-Gamma latent variables

The estimators we developed in the previous section allowed us to estimate the bound in an unbiased way, as long as the variational expectation over the likelihood only depended on $\mu_n$, $\mu_n^2$, and $\sigma_n^2$. This limits the applicability of our method to likelihoods of a Gaussian form. Luckily, some likelihoods can be written as expectations over unnormalised Gaussians. For example, the logistic likelihood can be written as an expectation over a Pòlya-Gamma variable $\omega$ [Polson et al., 2013]:

$$\sigma(y_n f(\mathbf{x}_n)) = (1 + \exp(-y_n f(\mathbf{x}_n)))^{-1} = \int c\mathcal{N}\left(f(\mathbf{x}_n); \frac{1}{2}y_n, \omega_n^{-1}\right)p(\omega_n)\mathrm{d}\omega_n\,. \qquad (22)$$

This trick was investigated by Gibbs and Mackay [2000] and recently revisited by Wenzel et al. [2018] to construct a variational lower bound to the logistic likelihood with a Gaussian form:

$$\log p(y_n|f(\mathbf{x}_n)) = \log\sigma(y_n f(\mathbf{x}_n)) \geq \mathbb{E}_{q(f(\mathbf{x}_n))q(\omega_n)}\left[c\log\mathcal{N}\left(f(\mathbf{x}_n)|\frac{1}{2}y_n, \omega_n^{-1}\right)\right]\,. \qquad (23)$$

This bound for the likelihood can be used as a drop-in approximation for the exact likelihood, at the cost of adding an additional KL gap to the true marginal likelihood. For our purpose, the crucial benefit is that we again obtain a Gaussian form in the expectation over $q(f(\mathbf{x}_n))$, for which we can use the unbiased estimators we developed above. Gibbs and Mackay [2000] and Wenzel et al. [2018] go on to find the optimal parameters for $q(\omega_n)$ in closed form. We cannot use this, as the optimal parameters depend non-linearly on $\mu_n$ and $\sigma_n$. Instead, we choose to parameterise $q(\omega_n)$ as a Pòlya-Gamma distribution with its parameters given by a recognition network mapping from the corresponding input and label, following Kingma and Welling [2014]. This method can likely be extended to the multi-class setting using the stick-breaking construction by Linderman et al. [2015].

## 5    Experiments

We demonstrate our approach on a series of experiments on variants of the MNIST datasets. While MNIST has been accurately solved by other methods, we intend to show that a model like an RBF GP (Radial Basis Function or squared exponential kernel), for which MNIST is challenging, can be significantly improved by learning the correct invariances. For binary classification tasks, we will use the Pòlya-Gamma approximation for the logistic likelihood, while for multi-class classification, we are currently forced to use the Gaussian likelihood. We consider two classes of transformations for which we automatically learn parameters: (i) global affine transformations, and (ii) local deformations. Note that we must be able to backpropagate through these transformations in order to learn their parameters.

**Affine transformations.**   2D affine transformations are determined by 6 parameters $\phi$ and allow for scaling, rotation, shear, and translation. To sample from $p(\mathbf{x}_a|\mathbf{x})$, we first draw $\phi \sim \text{Unif}(\phi_{\min}, \phi_{\max})$ and then apply[3] the transformation to obtain $\mathbf{x}_a = \text{Aff}_\phi(\mathbf{x})$. Since the transformation $\text{Aff}_\phi(\cdot)$ is differentiable w.r.t. $\phi$, we can backpropagate to $\{\phi_{\min}, \phi_{\max}\}$ using the reparameterisation trick.

**Local deformations.**   As a second class of transformations we consider the local deformations as introduced with the infiniteMNIST dataset [Loosli et al., 2007; Simard et al., 2000]. Samples are created by first creating a smooth vector field to describe the local deformations, followed by local rotations, scalings, and various other transformations. The parameters determining the size of the transformations can be backpropagated through in the same way as for the affine transformations.

### 5.1    Recovering invariances in binary MNIST classification

As a first test, we demonstrate that our approach can recover the parameter of a known transformation in an odds-vs-even MNIST binary classification problem. We consider the regular MNIST dataset and rotate each example by a randomly chosen angle $\phi \in [-\alpha_{\text{true}}, \alpha_{\text{true}}]$ for $\alpha_{\text{true}} \in \{90°, 180°\}$.

We choose $p(\mathbf{x}_a|\mathbf{x})$ to be a uniform distribution over rotated images, leading to a rotational invariance, and use the variational lower bound to train the amount of rotation $\alpha$. To perform well on this task, we expect the recovered $\alpha$ to be at least as large as the true value $\alpha_{\text{true}}$ to account for the rotational invariance. Too large values, i.e. $\alpha \approx 180°$, should be avoided due to ambiguities between, for example, 6s and 9s.

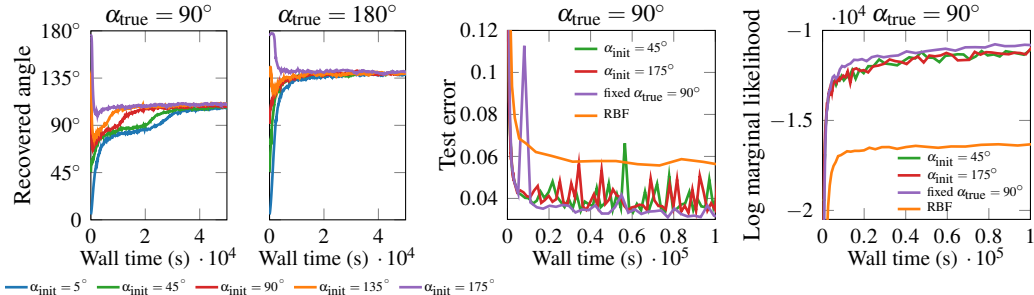

Figure 3:  Binary classification on the partially rotated (by $\pm 90°$ or $\pm 180°$) MNIST dataset. *Left:* Amount of rotation invariance over time. *Middle:* Test error. *Right:* Log marginal likelihood bound.

We find that the trained GP models with invariances are able to approximately recover the true angles (fig. 3, left). When $\alpha_{\text{true}} = 180°$, the angle is under-estimated, whereas $\alpha_{\text{true}} = 90°$ is recovered well. Regardless, all models outperform the simple RBF GP by a large margin, both in terms of error, and in terms of the marginal likelihood bound (fig. 3, right). These results show that our approach can be combined effectively with certain non-Gaussian likelihood models using the Pòlya-Gamma trick.

### 5.2    Classification of MNIST digits

Next, we consider full MNIST classification, using a Gaussian likelihood, and compare the non-invariant RBF kernel to various invariant kernels. Figure 4 shows that the GPs with invariant kernels clearly outperform the baseline RBF kernel. Both types of learned invariances, affine transformations and local deformations, lead to similar performance for a wide range of initial conditions. When constrained to rotational invariances only, the results are only slightly better than the baseline. This

indicates that deformations (stretching, shear) are more important than rotations for MNIST. Crucially, we do not require a validation set, but can use the log marginal likelihood of the training data to monitor performance. In fig. 1 we show samples from $p(\mathbf{x}_a|\mathbf{x})$ for the model that uses all affine transformations.

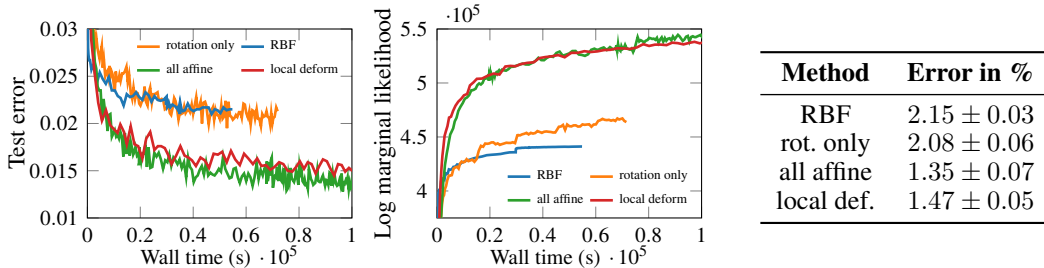

Figure 4: MNIST classification results. *Left:* Test error. *Middle:* Log marginal likelihood bound. *Right:* Final test error. All invariant models outperform the RBF baseline.

## 5.3 Classification of rotated MNIST digits

We also consider the fully rotated MNIST dataset[4]. In this case, we only run GP models that are invariant to affine transformations. We compare general affine transformations (learning all parameters), rotations with learned angle bounds, and fixed rotational invariance (fig. 5). We found that all invariant models outperform the baseline (RBF) by a large margin. However, the models with fixed angles (no free parameters of the transformation) outperform their learned counterparts. We attribute this to the optimisation dynamics, as the problem of optimising the variational, kernel, and transformation parameters jointly is more difficult than optimising only variational and kernel parameters for fixed transformations. We emphasise that the marginal likelihood bound does correctly identify the best-performing invariance in this case as well.

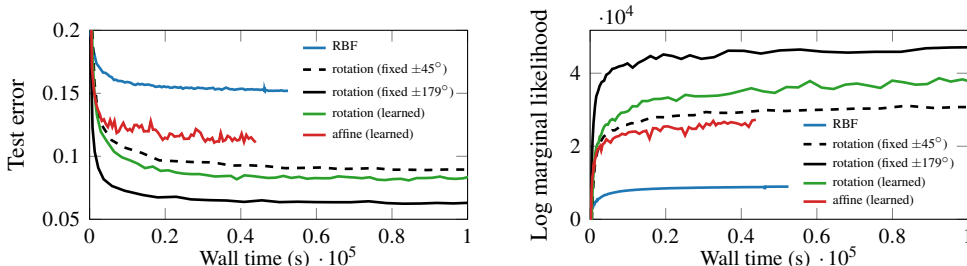

Figure 5: Rotated MNIST classification results. *Left:* Test error. *Right:* Log marginal likelihood bound. The optimiser has difficulty finding a good solution with the learned invariances, although the marginal likelihood bound correctly identifies the best model.

## 6 Conclusion

In this work, we show how invariances described by general data transformations can be incorporated into Gaussian process models. We use "double-sum" kernels, which sum a base kernel over all points that are similar under the invariance. These kernels cannot be evaluated in closed form, due to integrals or a prohibitively large number of terms in the sums. Our method solves this technical issue by constructing a variational lower bound which only requires unbiased estimates of the kernel. Crucially, this variational lower bound also allows us to learn a good invariance by maximising the marginal likelihood bound through backpropagation of the sampling procedure. We show experimentally that our method can learn useful invariances for variants of MNIST. In some experiments, the joint optimisation problem does not achieve the performance of when the method is initialised with the correct invariance. Despite this drawback, the objective function correctly identifies the best solution. While in this work we focus on kernels with invariances, we hope that our demonstration of learning with kernels that do not admit a closed-form evaluation will prove to be more generally useful by increasing the set of kernels with interesting generalisation properties that can be used.

**Acknowledgements**

M.B. gratefully acknowledges partial funding through a Qualcomm studentship in technology.

## Footnotes

[1]Conceptually, a neural network could be used if an accurate estimate of its marginal likelihood were available.

[2]Sub-sampling without replacement requires a different re-weighting of diagonal elements (appendix B).

[3]Affine transform implementation from `github.com/kevinzakka/spatial-transformer-network`.

[4] http://www.iro.umontreal.ca/~lisa/twiki/bin/view.cgi/Public/MnistVariations

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
