[Supplementary Material]

# Learning Invariances using the Marginal Likelihood: Appendix

**Mark van der Wilk**
PROWLER.io
Cambridge, UK
mark@prowler.io

**Matthias Bauer**
MPI for Intelligent Systems
University of Cambridge
msb55@cam.ac.uk

**ST John**
PROWLER.io
Cambridge, UK
st@prowler.io

**James Hensman**
PROWLER.io
Cambridge, UK
james@prowler.io

## A   Insensitivity

In the main text we described that we want to construct functions which are *insensitive* to certain input variations. We describe the variations for a particular input $\mathbf{x}$ using the distribution $p(\mathbf{x}_a|\mathbf{x})$, and we want to limit the probability of a large difference in the function $f(\cdot)$ as

$$P\Big([f(\mathbf{x}) - f(\mathbf{x}_a)]^2 > L\Big) < \epsilon \qquad \forall \mathbf{x} \in \mathcal{X} \qquad \mathbf{x}_a \sim p(\mathbf{x}_a|\mathbf{x}) \,. \tag{1}$$

For a given $\epsilon$, a smaller $L$ implies more insensitivity. In this note, we aim to quantify the degree of insensitivity for priors with double sum kernels. To simplify our analysis, we assume that $p(\mathbf{x}_a|\mathbf{x})$ randomly samples points from an "augmentation set" $\mathcal{A}(\mathbf{x})$, and that our kernel is defined as

$$k(\mathbf{x}, \mathbf{x}') = \frac{\sum_{\mathbf{x}_a \in \mathcal{A}(\mathbf{x})} \sum_{\mathbf{x}'_a \in \mathcal{A}(\mathbf{x}')} k_g(\mathbf{x}_a, \mathbf{x}'_a)}{\sqrt{\sum_{\mathbf{x}_a, \mathbf{x}_b \in \mathcal{A}(\mathbf{x})} k(\mathbf{x}_a, \mathbf{x}_b)} \sqrt{\sum_{\mathbf{x}'_a, \mathbf{x}'_b \in \mathcal{A}(\mathbf{x}')} k(\mathbf{x}'_a, \mathbf{x}'_b)}} \,, \tag{2}$$

i.e. we normalise the double sum kernel from the main paper. We do this to ensure that we always retain a unit marginal variance $k(\mathbf{x}, \mathbf{x}')$, in order to ensure that our kernel can actually learn something. Without this constraint, we can trivially pick $\mathcal{A}(\mathbf{x})$ to average many distant points, which makes all $f(\cdot)$s insensitive, but also completely constant. We do not need to apply this constraint in our practical method, as we choose the scale of the kernel by optimising the marginal likelihood.

We start by bounding the deviation of functions under the prior between $\mathbf{x}$ and $\mathbf{x}_a$.

**Lemma.** *When $f \sim \mathcal{GP}(0, k(\mathbf{x}, \mathbf{x}'))$ we can bound the probability of a deviation as*

$$P\Big([f(\mathbf{x}) - f(\mathbf{x}_a)]^2 > 2\epsilon(1 - \mathbb{E}_{\mathbf{x}_a|\mathbf{x}}[k(\mathbf{x}, \mathbf{x}_a)])\Big) < \frac{1}{\epsilon} \,. \tag{3}$$

*Proof.* We apply Markov's inequality to the random variable $[f(\mathbf{x}) - f(\mathbf{x}_a)]^2$:

$$P\Big([f(\mathbf{x}) - f(\mathbf{x}_a)]^2 > \epsilon \mathbb{E}_{f, \mathbf{x}_a|\mathbf{x}}\Big[[f(\mathbf{x}) - f(\mathbf{x}_a)]^2\Big]\Big) < \frac{1}{\epsilon} \,. \tag{4}$$

The expectation over $f(\cdot)$ evaluates as

$$\mathbb{E}_f\Big[[f(\mathbf{x}) - f(\mathbf{x}_a)]^2\Big] = \mathbb{E}_f\big[f(\mathbf{x})^2 - 2f(\mathbf{x})f(\mathbf{x}_a) + f(\mathbf{x}_a)^2\big] = k(\mathbf{x}, \mathbf{x}) - 2k(\mathbf{x}, \mathbf{x}_a) + k(\mathbf{x}_a, \mathbf{x}_a)$$
$$= 2(1 - k(\mathbf{x}, \mathbf{x}_a)) \,, \tag{5}$$

leaving only the expectation over $\mathbf{x}_a$ as in the statement. $\qquad \square$

This shows that we can increase the insensitivity of functions in the prior by increasing $\mathbb{E}_{\mathbf{x}_a|\mathbf{x}}[k(\mathbf{x}, \mathbf{x}_a)]$. Not all distributions $p(\mathbf{x}_a|\mathbf{x})$ actually increase the expected covariance. In our method, we simply parameterise kernels that allow insensitivity, and then let the marginal likelihood determine the extent to which it is applied. As an example, we take $k_g(\mathbf{x}, \mathbf{x}')$ to be a squared exponential kernel. We can increase the expected covariance by ensuring that the augmentation densities of $\mathbf{x}$ and a random augmented point $\mathbf{x}_a$ overlap largely. If the augmentation distributions fully and uniformly overlap, we obtain strict invariance again.

# B Unbiased estimators for kernel matrices

Here, we elaborate on the derivation of unbiased estimators for the kernel expressions that are needed to compute the variational lower bound. In addition to the estimator for double-integral kernels, we will also discuss estimators for double-sum kernels more closely.

## B.1 Double-sum kernels

When we consider strictly invariant kernels or $p(\mathbf{x}_a | \mathbf{x})$ with discrete support, the kernel becomes a sum over the augmentation set $\mathcal{A}(\mathbf{x})$ of size $P$. Our kernel and inducing variable covariances are:

$$k_f(\mathbf{x}, \mathbf{x}') = \sum_{\mathbf{x}_a \in \mathcal{A}(\mathbf{x})} \sum_{\mathbf{x}'_a \in \mathcal{A}(\mathbf{x})} k_g(\mathbf{x}_a, \mathbf{x}'_a), \qquad k_{\mathbf{fu}}(\mathbf{x}, \mathbf{z}) = \sum_{\mathbf{x}_a \in \mathcal{A}(\mathbf{x})} k_g(\mathbf{x}_a, \mathbf{z}). \tag{6}$$

We aim to find unbiased estimates for the squared predictive mean $\mu_n^2 = (\mathbf{k}_{\mathbf{f}_n \mathbf{u}} \mathbf{K}_{\mathbf{uu}}^{-1} \mathbf{m})^2$, and the predictive covariance $\sigma_n^2 = k_f(\mathbf{x}_n, \mathbf{x}_n) - \mathbf{k}_{\mathbf{f}_n \mathbf{u}} \mathbf{K}_{\mathbf{uu}}^{-1}(\mathbf{K}_{\mathbf{uu}} - \mathbf{S}) \mathbf{K}_{\mathbf{uu}}^{-1} \mathbf{k}_{\mathbf{uf}_n}$, for which we require

$$I = \sum_{\mathbf{x}_a \in \mathcal{A}(\mathbf{x})} \sum_{\mathbf{x}'_a \in \mathcal{A}(\mathbf{x})} r(\mathbf{x}_a, \mathbf{x}'_a) \tag{7}$$

for $r = k_f(\mathbf{x}_a, \mathbf{x}'_a)$ and $r = k_{\mathbf{fu}}(\mathbf{x}_a, \mathbf{z}_m) k_{\mathbf{fu}}(\mathbf{x}'_a, \mathbf{z}_{m'})$. As mentioned in the main text, the most obvious unbiased estimator would sub-sample two different sets of $\mathcal{A}$ for each of the two sums. However, given the cost of transforming input images, we aim to use the *same* subset for both sums. This additionally speeds up the kernel computations, as the same covariances with the inducing points are needed. We also want to sample *without* replacement to reduce variance. We denote the randomly sampled subset as $\mathcal{M} \subset \mathcal{A}(\mathbf{x})$, and for now denote its elements as $\mathbf{x}_i$, with $i \leq M$. In order to ensure uniform weighting over all elements in the sum, we re-weight the diagonal elements in the estimator:

$$\hat{I} = \sum_{i=1}^{M} \sum_{j=1}^{M} r(\mathbf{x}_i, \mathbf{x}_j) \left( \frac{P(P-1)}{M(M-1)}(1 - \delta_{ij}) + \frac{P}{M} \delta_{ij} \right). \tag{8}$$

We show that this estimator is unbiased by taking an expectation over the random subset $\mathcal{M}$. To simplify expressions, we first separate the sum in eq. (7) into the off- and on-diagonal components:

$$I = \sum_{\mathbf{x}_a \in \mathcal{A}(\mathbf{x})} \sum_{\substack{\mathbf{x}'_a \in \mathcal{A}(\mathbf{x}) \\ \mathbf{x}'_a \neq \mathbf{x}_a}} r(\mathbf{x}_a, \mathbf{x}'_a) + \sum_{\mathbf{x}_a \in \mathcal{A}(\mathbf{x})} r(\mathbf{x}_a, \mathbf{x}_a) = I_{\neg d} + I_d. \tag{9}$$

Remember that the distribution $p(\mathbf{x}_i, \mathbf{x}_j)$ samples without replacement, and for $i = j$ the density equals the marginal $p(\mathbf{x}_i)$. We summarise this as

$$p(\mathbf{x}_i = \mathbf{x}_a, \mathbf{x}_j = \mathbf{x}'_a) = (1 - \delta(\mathbf{x}_a - \mathbf{x}'_a))(1 - \delta_{ij}) \frac{1}{P(P-1)} + \delta_{ij} \delta(\mathbf{x}_a - \mathbf{x}'_a) \frac{1}{P} \tag{10}$$

We now take the expectation over $\hat{I}$:

$$\mathbb{E}_{p(\mathcal{M})}\left[\hat{I}\right] = \sum_{i=1}^{M} \sum_{j=1}^{M} \sum_{\mathbf{x}_a \in \mathcal{A}(\mathbf{x})} \sum_{\mathbf{x}'_a \in \mathcal{A}(\mathbf{x})} p(\mathbf{x}_i = \mathbf{x}_a, \mathbf{x}_j = \mathbf{x}'_a) r(\mathbf{x}_i, \mathbf{x}_j) w_{ij}$$

$$= \sum_{i=1}^{M} \sum_{j=1}^{M} (1 - \delta_{ij}) \frac{w_{ij}}{P(P-1)} I_{\neg d} + \delta_{ij} \frac{w_{ij}}{P} I_d$$

$$= \frac{P(P-1)}{M(M-1)} \frac{M(M-1)}{P(P-1)} I_{\neg d} + \frac{P}{M} \frac{M}{P} I_d = I. \tag{11}$$

For ease of implementation, we re-write the estimator in terms of a full and diagonal sum over $r$:

$$\hat{I} = \frac{P(P-1)}{M(M-1)} \sum_{i=1}^{M} \sum_{j=1}^{M} r(\mathbf{x}_i, \mathbf{x}_j) + \frac{P(M-P)}{M(M-1)} \sum_{i=1}^{M} r(\mathbf{x}_i, \mathbf{x}_i). \tag{12}$$

## B.2 Double-integral kernels

We now consider double integral kernels, giving kernel and inducing variable covariances of:

$$k_f(\mathbf{x}, \mathbf{x}') = \iint p(\mathbf{x}_a|\mathbf{x})p(\mathbf{x}'_a|\mathbf{x}')k_g(\mathbf{x}_a, \mathbf{x}'_a)\mathrm{d}\mathbf{x}_a\mathrm{d}\mathbf{x}'_a \,, \tag{13}$$

$$k_{\mathbf{fu}}(\mathbf{x}, \mathbf{z}) = \int p(\mathbf{x}_a|\mathbf{x})k_g(\mathbf{x}_a, \mathbf{z})\mathrm{d}\mathbf{x}_a \,. \tag{14}$$

Informally, we can think of this as the double-sum kernel, where we take a limit to an infinite augmentation set and divide by $P^2$ to normalise. This gives us the estimator from the main paper:

$$\hat{I} = \frac{1}{M(M-1)}\left[\sum_{i=1}^{M}\sum_{j=1}^{M} r(\mathbf{x}_i, \mathbf{x}_j) - \sum_{i=1}^{M} r(\mathbf{x}_i, \mathbf{x}_i)\right], \qquad \mathbf{x}_i \sim p(\mathbf{x}_a|\mathbf{x}). \tag{15}$$

We can show it to be unbiased by taking the expectation over all $\mathbf{x}_i$, and a small re-arrangement:

$$\mathbb{E}_{p\left(\{\mathbf{x}_i\}_{i=1}^{M}|\mathbf{x}\right)}\left[\hat{I}\right] = \frac{1}{M(M-1)}\sum_{i=1}^{M}\sum_{j=1}^{M}(1 - \delta_{ij})\mathbb{E}[r(\mathbf{x}_i, \mathbf{x}_j)]$$

$$= \iint p(\mathbf{x}_i|\mathbf{x})p(\mathbf{x}_j|\mathbf{x})r(\mathbf{x}_i, \mathbf{x}_j)\mathrm{d}\mathbf{x}_i\mathrm{d}\mathbf{x}_j = I \tag{16}$$

### B.2.1 Computational complexity

The estimator for $k_f(\mathbf{x}, \mathbf{x}')$ costs $\mathcal{O}(M^2)$, as the double sum needs to be evaluated for each element. When $r = k_{\mathbf{fu}}(\mathbf{x}_a, \mathbf{z}_m)k_{\mathbf{fu}}(\mathbf{x}'_a, \mathbf{z}_{m'})$, the estimator costs only $\mathcal{O}(M)$, as the double sum factorises over each term.