[Reviews · NeurIPS 2018]

Reviewer 1



In this manuscript the authors present a scheme for ranking and refining invariance transformations within the Bayesian approach to supervised learning, thereby offering a fully Bayesian alternative to data augmentation. The implementation presented also introduces a scheme for unbiased estimation of the evidence lower bound for transformation+representation models having a Normal likelihood (or writeable in an equivalent form as per the Polya-Gamma example). Moreover, the authors have been careful to ensure that their implementation is structured for efficiency within the contemporary framework for stochastic optimisation of sparse approximations to the GP (variational inference with mini-batching, etc.). Although limited by space, the examples presented are convincing of the potential of this approach. It is my view that this is a valuable and substantial contribution to the field, although I would be prepared to concede in relation to the NIPS reviewer guidelines that in some sense the progress is incremental rather than revolutionary. The primary concerns I have regarding the proposed method are common to: (i) the entire family of variational methods: namely, the limited justification for use of the evidence lower bound as approximation to the marginal likelihood for model selection purposes; and (ii) the entire process of model selection by marginal likelihood: the role of prior-sensitivity. That is to say I would be very interested to see a comparison between complete and variational treatments of this problem one a class of models over which the complete Bayesian approach is computationally feasible. Likewise, to see a connection against existing methods for complete marginal likelihood estimation over group-structured spaces (e.g Kong et al. JRSSB, 2003, 65:585-618). But this is well beyond the scope of a single NIPS contribution. Typos: p5: "stems less the bound" ? p8: "double-sum over the of the" ? Minor: p2. "We believe this work to be exciting, as it simultaneously provides a justification for data augmentation from a Bayesian point of view" ... seems to me more like this manuscript is an argument *against* doing data augmentation ?

Reviewer 2



Summary This paper proposes an invariant Gaussian process as prior for regression and classification models. A variational inference shame is derived and the invariance is described by a probability distribution that can be sampled from. Strength The authors provided a relatively new perspective on how the model should be robust to the minor changes of the input variable values. Weakness The logical flow and technical presentation are very confusing, which is the biggest problem of this paper. First, in Section 2, the authors introduced the definition of a strong and weak form of invariance. If such invariance is already accepted and studied, the authors should have pointed out previous works. Otherwise, the authors should have explained many details, such as the range of transformation T, i.e., what kind of transformations should be considered. Also, why the invariance is defined through Equation (1) and (2). Are there any other proper form of definitions? Second, the authors confused many things, including the purpose of regularization and the purpose of making the model robust to input. So the review on Regularization and Model Constraints in Section 2 is not necessary. Third, in the most important part, Section 4, the authors introduced too many concepts and notation without a proper definition. In Section 4.1, what should be the set of G? What is the meaning of orbit? It is directly related to how many Gaussian processes are being summed in Equation (4). In Section 4.4, the entire notation, especially x_a and x_n, x_a’ are used without definition. Overall, the quality of this paper is hard for me to assess, due to its poor technical development and presentation. I am not convinced that this work contains some original idea and has any kind of significance to the supervise learning community.

Reviewer 3



The paper deals with incorporating invariances in Gaussian processes. Specifically, the authors propose a method for making Gaussian process models less sensitive to transformations (affine transformations, rotations etc). In their set-up, the transformations under consideration are described by a probability distribution and the invariance is achieve by averaging the kernel of the Gaussian process wrt. this distribution of transformations. For this class of models, both the kernel of invariant functions as well as the posterior distributions of interest are intractable. However, the authors deal with both of these issues using the so-called inter-domain inducing points approach. One major advantage of the proposed method is that the only requirement for the distribution of the transformations is that it can be sampled from. Importantly, the proposed method also allows learning parameters of the transformations from data through the marginal likelihood. The drawback is that they are only able to derive lower bounds for the marginal likelihood of the invariant model for Gaussian likelihoods and logistic likelihoods. The paper is concluded by applying the proposed method to the MNIST dataset and some of its variations. The submission appears technically sound and correct. The claims of the paper are not supported by any theoretical analysis, but the paper does contain experimental results for the proposed method showing the benefits of incorporating invariance. My prime concern is that the only included datasets are MNIST and variations of MNIST and yet, their results still seem quite sensitive. It would be a significant improvement, if the authors could demonstrate the benefit of the model on more complex datasets. Also, it would be interesting if the authors would compare their method against data augmentation. The paper is in general well-structured, well-written and easy to follow. Their idea for incorporating invariance in Gaussian processes in original and novel. Incorporating invariance into probabilistic models is an important problem. Despite the fact that experimental results could be more convincing and it is not possible to do multi-class classification in a proper probabilistic fashion, I still believe that the results are important and that other researchers will most likely build upon and improve their method. Further comments: Line 26: “...likelihood cannot be used to compare to other model.” typo Line 23, 36: The following two sentences are contradictions: “Data augmentation is also unsatisfactory from a Bayesian perspective” (line 23) and “We believe this work to be exciting, as it simultaneously provides a justification for data augmentation from a Bayesian point of view” (line 36) Line 134: “Data augmentation attempts to do the same, only by considering a general set points for each input x.” typo Line 189: “The intractability stems less the bound than the approximate posterior q(f ), …” Something seems to be missing here Line 228: How many samples S is needed in practice? Line 234: “For example, the logistic likelihood can be written as an expectation over a Pòlya-Gamma variable ω [REF]”. Missing reference Line 275: “However, all invariant models outperform a GP model with simple RBF kernel by a large margin both in terms of error and marginal likelihood (not shown), see Fig. 2 (right).” If the right-most panel does not show the marginal likelihood, then what does it show? After author feedback ----------------------------- I've read the rebuttal, but it has not changed my opinion: it is a good submission with significant contributions to the field.